# Novel Deployable Panel Structure Integrated with Thick Origami and Morphing Bistable Composite Structures

**DOI:** 10.3390/ma15051942

**Published:** 2022-03-05

**Authors:** Shuyong Ding, Min Sun, Yang Li, Weili Ma, Zheng Zhang

**Affiliations:** 1Zhijiang College of Zhejiang University of Technology, Shaoxing 312030, China; dsy@zjc.zjut.edu.cn; 2College of Mechanical Engineering, Zhejiang University of Technology, Hangzhou 310014, China; sunmin@zjut.edu.cn (M.S.); 18989492565@163.com (Y.L.); 15957178549@163.com (W.M.); 3Key Laboratory of Special Purpose Equipment and Advanced Processing Technology, Ministry of Education and Zhejiang Province, Zhejiang University of Technology, Hangzhou 310014, China

**Keywords:** origami structure, deployable bistable structure, finite element analysis, mechanical testing, assembly

## Abstract

This paper proposes a novel deployable panel structure integrated with a bistable composite structure and thick panel based on the thick origami technique. To overcome the interference effects between thick panels, the axis shift method is used in this deployable structure design. Bistable composite structures are employed as hinges for morphing characteristics. The trigger force and load-displacement curves of the structure are obtained by experiments and numerical simulations. The factors that affect the coverage area-to-package volume ratio and trigger force are discussed. The experimental and numerical results verify that the structure has two stable configurations and a large coverage area-to-package volume ratio.

## 1. Introduction

A deployable structure has the capacity to transform and predictably adopt multiple predetermined configurations, moving through known paths while deploying in a controlled and safe way [1]. Deployable structures have been defined as convertible structures with the capacity to undergo large configuration changes in an autonomous manner [2], and the reverse process is called retraction. There are some deployable structures that can maintain static equilibrium at each stage of deployment [3,4], thereby offering an even greater range of adaptability. The shape, properties and behavior of deployable structures can be changed to adapt to their surroundings and specific application environments [5]. A deployable structure is a mobile assembly that does not target motion but attains different configurations depending on the service requirements. Deployable structures can use mechanisms to achieve their changing circumstances, such as their shape or function [6]. At a basic level, a deployable structure is a kind of spatial structure formed by plates or sticks based on different configurations in space [7]. Deployment refers to the transformation of these structures from small, tight and compact configurations to unfolded and open configurations with load-carrying capacity [8,9]. These structures are primarily used for transportation purposes or applications requiring shape adaptability, such as a solar panel [10], which allows more panels to be in direct sunlight at any given point, facilitating power generation.

Origami is the ancient art of folding paper, where a two-dimensional structure is realized into complex, three-dimensional configurations. Origami technology has become a research hotspot in the aerospace [11], bionic structure [12,13,14] and microrobotics [15,16] fields. Many novel and practical folding methods have been designed and developed, among which the Miura pattern is the most pioneering and common folding method [17,18]. A comprehensive experimental and analytical study on the dynamics of origami folding by investigating a stacked Miura origami structure with intrinsic bistability has been presented by Fang et al. [19]. This study provides fundamental insights into the dynamics of origami folding and provides a solid foundation for developing foldable and deployable structures. Apart from the above methods, there are several more common folding methods, such as waterbomb [20,21], which is a single-vertex, bistable mechanism with unique properties for smart materials and control systems. Hanna et al. [22] report a quantitative investigation of the kinetic behavior of a symmetric waterbomb base. The multiforming of the origami structure can be obtained by folding the panel along the crease; nevertheless, the state of the origami structure is unstable when the deformation is completed or the stability tends to be destroyed. Chillara and Dapino [23] propose a strategy for the creation of smooth folds in finite-thickness laminated composites; this approach is applicable to smart, folding structures with reconfigurable creases inspired by origami technology.

A bistable composite structure has two different stable configurations without the need for a continuous external energy supply [24,25,26]. This smart structure has great practical value and wide application in deployable structures [27,28], mechanical engineering [29,30] and even biological engineering fields [31,32]. Generally, bistable composite structures can be divided into two categories according to their stacking sequence. The first one is named the cross-ply composite structure [33,34,35], which has been comprehensively studied. The effect of the initial curvature of the tool plate on the snap-through load of a square cross-ply bistable composite was analyzed in a previous study [36]. A cross-ply bistable composite structure transforms between two stable states once subjected to a small given load, which means it has a low load-carrying capacity. Therefore, another bistable composite structure, named the anti-symmetric composite structure [37,38], has been proposed. The main characteristics of anti-symmetric layup that differ from the cross-ply layup are as follows: (1) the force to trigger the snap-through and snap-back processes of the anti-symmetric structure is higher than that of the cross-ply structure; (2) the two different configurations of the cross-ply structure have opposite directions, while the directions of two stable shapes for anti-symmetric structures are the same. Zhang et al. [37,39] comprehensively investigate this novel structure by considering temperature effects [37], hygroscopic influences and viscoelastic properties. As a stable deployable panel structure is expected, anti-symmetric layups have been selected in this study.

This paper presents a deployable panel structure strategy by combining the bistable structure and thick panel. The deployable structure is proposed based on the principle of thick origami for specimen manufacturing and numerical simulation. The specimens of the deployable panel structure have been designed and manufactured according to the thick origami technique, and experiments have been conducted to investigate the trigger force and load–displacement curves. Finally, finite element (FE) simulation of the morphing process has been conducted by using ABAQUS software. The results of the numerical simulation and experimental test are compared and found to be in good agreement.

## 2. Design of the Deployable Panel Structure

In this section, a deployable panel with bistable anti-symmetric carbon fiber-reinforced polymer (CFRP) shells and a thick origami technique is designed and assembled, as shown in Figure 1a,b, which includes the design scheme, to solve the folding problem caused by a thick panel.

As a deployable structure, the panel model should have a certain capability to change in volume and surface area. As shown in Figure 1c,d, the deployable model mainly includes two parts: the deployable panel and the control unit (anti-symmetric CFRP cylindrical shell here). Inspired by the origami [40] technique, the movement of the panel has been designed to rotate around the crease. As the thickness of the panel increases, interference effects are generated during the folding of the structure. Several solutions, such as the tapered panel technique, offset panel technique, hinge shift technique and rolling contacts technique, have been developed to overcome this problem. The axis shift method is adopted in this paper because the integrity of the deployable panel model can be guaranteed in this situation.

An overlapping region occurs for deployable structures with thick panels, and it is crucial to address this issue before these structures can be realized as deformable or morphing structures. In terms of the appearance and integrity of the structure, the axis shift method is used to make the deployable structure with thick panels. To prevent collisions between the panels, each rotational axis is shifted to the top of the thick panel instead of the mid-surface as the thickness increases. In this case, the thick panel can only be folded along a fixed direction; therefore, a novel structure with one drilling degree of freedom is required as a rotational axis. The bistable CFRP cylindrical shell is employed as the hinge to realize the folding action of the entire model.

The model transforms from the folded configuration to the flattened configuration under the action of external loading. The magnitude of the required external force to drive the model should be determined. As this model is symmetric in terms of the structure, one quarter of the model is selected as the research object, as shown in Figure 1e,f. To investigate the mechanical properties of this deployable model, experiments and finite element simulations are carried out. The geometric parameters and material parameters of the deployable panel model are given in Table 1 and Table 2, respectively. The central angle of the cylindrical shell is denoted as *β*, *L*_c_ is the length of the cylindrical shell, *R* is the radius of the cylindrical shell and *L*_j_ is the distance between two junctions. *L*_s_ is the sidelength of the deployable panel, and the thicknesses of the deployable panel and cylindrical shell are labeled *h* and *t*, respectively.

The origami technology used in this article can convert two-dimensional planar structures into three-dimensional structures, accompanied by large changes in surface area and contained volume. In recent years, it has been deeply researched, attracting great attention in the fields of deformable structure, energy collection, architecture and even bionics. In this paper, a novel solar panel structure is investigated by combining bistable shells with origami techniques.

## 3. Experimental Test and FE Simulation

Experimental tests and corresponding numerical simulations were performed on the deforming process of the deployable panel model in this section. The trigger force required for the stable transformation of the deployable structure was measured through experiments and the finite element software ABAQUS.

### 3.1. Measurement of the Trigger Force

To investigate the mechanical properties of the deployable panel model by the experimental method, a prototype of the model was designed and manufactured. The components of the deployable panel model were assembled according to the three-dimensional model, as shown in Figure 1. As discussed in the former section, a bistable CFRP cylindrical shell was selected as the actuator and control unit of the deployable structure. The stacking sequence of the CFRP cylindrical shell was [45°/−45°/45°/−45°]. The CFRP cylindrical shell and the panel were bonded together by using the screw connection shown in Figure 2. Cardboard sheets with a thickness of 3 mm were chosen as the rigid panels and bolted together. The cylindrical shells were fabricated in a cylindrical steel mold, cured at a high temperature 180 °C and then cooled to room temperature. The deployable panel model can maintain a flattened state and a cubic configuration in the folded state without an ongoing external energy supply.

The deformation process and trigger force of the deployable panel specimen were experimentally observed and measured by the universal tensile testing machine, as shown in Figure 3. The specimen was placed on the support, and the indenter moved downward. The related parameters of the universal tensile testing machine are listed as follows: (1) because the trigger force of the deployable structure is small [39], the force sensor with a measurement range of 0~50 N was selected; (2) the accuracy of the testing machine was less than 0.5%. Interruption conditions should be established to guarantee the safety of the equipment because the second configuration of the model was a planar structure and could lead to overload. Thus, loading would be interrupted under the following conditions: (1) an abrupt decrease in the reaction force is captured, which means that the snap-through process is completed; (2) the reaction force exceeds 10 N. Moreover, the loading speed in this experiment was 5 mm/min. The load–displacement relationship could be captured and displayed in the monitor.

The preparation of the cylindrical shell adopted the traditional autoclave process. First, the carbon fiber prepreg was laid on the cylindrical mold according to a specific layup method, and then the entire mold was packaged in a vacuum bag to ensure that the carbon fiber did not come into contact with air during high-temperature and high-pressure curing in an autoclave. The mold was solidified under high temperature and high pressure for three hours and then cooled to room temperature to obtain a bistable cylindrical shell. The critical load of the solar panel was tested by the Instron tensile testing machine

### 3.2. Numerical Simulation

In this section, numerical simulation of the deformation of the deployable panel model was conducted by using the finite element software ABAQUS, version 6.13. The finite element model was composed of three core components: (1) the panel (created as a rigid part), (2) the CFRP cylindrical shell (created by the composite option) and (3) the smooth support panel (created as a rigid part). The material properties of the cylindrical shell are given in Table 1. The geometric parameters of the CFRP cylindrical shell and deployable panel are listed in Table 2. Figure 4 shows the assembly relation of one quarter of the deployable panel model. The CFRP cylindrical shell was bound to the deployable panel at the junctions by the *Tie* option. Hard contact was established between the panel and CFRP cylindrical shell. This relationship was also applied to the interface between the support panel and the panel of the deployable structure. Two static steps were established: the loading step and the unloading step. To obtain a convergent solution, both the *Nlgeom* and *autostabilization* options were turned on. A displacement load was applied on the top edges of the deployable panel model in the loading step, which was then withdrawn in the unloading step. The corresponding boundary conditions are given as follows: six degrees of freedom of the support panel were constrained. The FE model of the shell was meshed by 1100 S4R elements.

Figure 5 shows the deformation process of the deployable panel model. Figure 5a indicates that the model is in the folded state; the model begins to deform as shown in Figure 5b; the configuration of the model in Figure 5c is unstable and ready to snap-through; Figure 5d shows that the snap-through process is completed and the model is in the second stable configuration; as shown in Figure 5e, to obtain a convergent solution, the position of the loaded edge exceeds the location where the second stable state is completed. The model began to recover under the effect of the CFRP cylindrical shell when the displacement load was withdrawn. Figure 5f shows that the model could remain in the second stable configuration without a sustained external energy supply.

Figure 6 shows the deformation process of the entire deployable panel, in which the two stable configurations show good agreement with experimental observations. The simulation procedure for the entire deployable model was similar to that of the one-quarter model. Figure 6a illustrates the entire deployable panel model in the folded state. Figure 6b shows the entire deployable panel model in an unfolded state, but the model is still loaded. In the unstable state shown in Figure 6c, the position of the loaded edge exceeds the location where the second stable state is completed. As the displacement load was withdrawn, the entire deployable panel remained in the second stable configuration without a sustained external energy supply, as shown in Figure 6d. The simulation results showed that the deployable panel model was able to transform between the folded and flattened states, with dramatic changes in the surface area and volume.

The bistable characteristics of the structure were mainly derived from the bistable characteristics of the cylindrical shell, which existed due to the inconsistent thermal expansion coefficients caused by the inconsistent layup angles of each layer of carbon fibers. According to previous simulation and experimental experience, material thickness has the greatest influence on the critical load. The structure discussed in this paper needed to have greater stiffness and stability in the unfolded state. In order to make the structure less prone to deformation, we aimed to maximize the critical load in the design.

## 4. Results and Discussion

### 4.1. Mechanical Properties of the Deployable Panel Model

Load–displacement curves reflect the mechanical properties of the deployable panel model under the action of external loading. Figure 7 shows the load–displacement curves obtained from the experiment and simulation. Both curves show that the load first increases with increasing displacement, followed by a sudden downward trend at the end of loading. This is because the snap-through process was completed and the model had transformed into the second stable configuration. After the load in the simulation curve was reduced to zero, it began to increase again. This was because the displacement load was still applied on the top edge of the model, as shown in Figure 5e. The experimental curve ends without reaching the value of zero because the interrupt condition was triggered.

The trigger force obtained in the experiment was 2.36 N, while the simulation value was 1.81. The trigger force of the experimental result was larger than that of the simulation result because partial energy was used to overcome the friction force between the CFRP cylindrical shell and the panel, and the accuracy of specimen manufacturing and assembly also contributed to the error. Another reason is that the material properties of the deployable panel model in different analysis methods may not have been strictly consistent.

### 4.2. Effect of the Distance between Two Junctions L_j_

The influence of the distance between two junctions, *L*_j_, when the deployable structure panels are in a flattened configuration, on the deployable models is discussed. Figure 8 shows two different stable configurations of the deployable panel model with different distances between two junctions *L*_j_. Affected by the screw bolt, the maximum value of *L*_j_ is 39.5 mm. Five cases of *L*_j_ = 20, 25, 30, 35, 39.5 mm are considered in this example. As the distance between two junctions *L*_j_ decreases, the greater of the two panels are unfolded. Table 3 shows that the coverage area-to-package volume ratio first decreased and then increased as the distance between the two junctions increased. The model had the largest coverage area-to-package volume ratio when the two plates were perpendicular, that is, the distance between two junctions *L*_j_ was 39.5 mm.

### 4.3. Effect of the Sidelength of the Deployable Panel L_s_

In this section, the influence of the sidelength of the deployable panel *L*_s_ on the trigger force of the deployable panel is discussed. Figure 9 shows two different stable configurations of the deployable panel model with different panel dimensions. Five cases of *L*_s_ = 70, …, 110 mm are considered in this paper. The corresponding load–displacement curves for different values of *L*_s_ are presented in Figure 10. This shows that the trigger force of the deployable panel model decreased with increasing sidelength *L*_s_. It should be noted that the force decreased dramatically to zero at the end, which means that the second stable state was obtained.

The deployable structure specimens with different panel sidelengths were manufactured as shown in Figure 9a. The load–displacement curves of each specimen were obtained by a compression test on the universal tensile testing machine and are shown in Figure 11. The trigger forces measured in the experiments were also greater than those of the FE simulations. Table 4 presents the main technical parameters of the deployable models with different panel sidelengths. The coverage area-to-package volume ratio decreased as the panel sidelength increased.

### 4.4. Effect of the Number of Layers of Bistable CFRP Cylindrical Shells

The influence of the geometric parameters of bistable CFRP cylindrical shells was also studied. The stacking sequences of the cylindrical shell were [45°/−45°]_2,_ [45°/−45°/0°/45°/−45°], [45°/−45°]_3,_ [45°/−45°/45°/0°/−45°/45°/−45°] and [45°/−45°]_4_. Figure 12 demonstrates the influence of the number of layers of the CFRP cylindrical shell on the load–displacement curves. The results show that the trigger force of the deployable panel model increased significantly with an increase in the number of layers. This was because the bistable CFRP cylindrical shell with more layers had a higher stiffness and consequently required a greater force for the transition between the two stable states.

Therefore, deployable panel models with different coverage area-to-package volume ratios and load-carrying capacities could be designed by varying the panel dimensions and geometric parameters of bistable CFRP cylindrical shells to meet the mechanical needs under various working conditions.

## 5. Conclusions

This paper presents a deployable panel structure inspired by the thick origami structure and bistable composite cylindrical shell that can remain in either a folded state (spatial) or a flattened (unfolded) state without the need for continuous external loading. The structural design of the model was given. Specimens were fabricated and assembled according to the geometric parameters based on thick origami technology and the axis shift method. In addition, finite element simulations were carried out and compared with experimental results. The error between the finite element simulation and the experimental results is less than 10%. The factors affecting the deployable structure’s trigger force and load-displacement curve have been discussed. The trigger force of the deployable structure increases significantly as the number of layers increases and decreases with increasing panel sidelength. The coverage area-to-package volume ratio decreases first and then increases as the distance between two junctions increases, and the coverage area-to-package volume ratio decreases with increasing panel sidelength. Both experimental and numerical results show that the proposed deployable panel model can remain either a folded (spatial) state or an unfolded (flatten) form, which sheds light on the design of space deployable structures. The deployable panel structure has a large coverage area-to-package volume ratio, which shows its application potential for solar panels. The application of this structure in solar panels is not studied in this article. In future research we will propose an overall mechanism for a feasible solar panel with this structure, and larger size cylindrical shells and plates will be considered.

## Figures and Tables

**Figure 1 materials-15-01942-f001:**
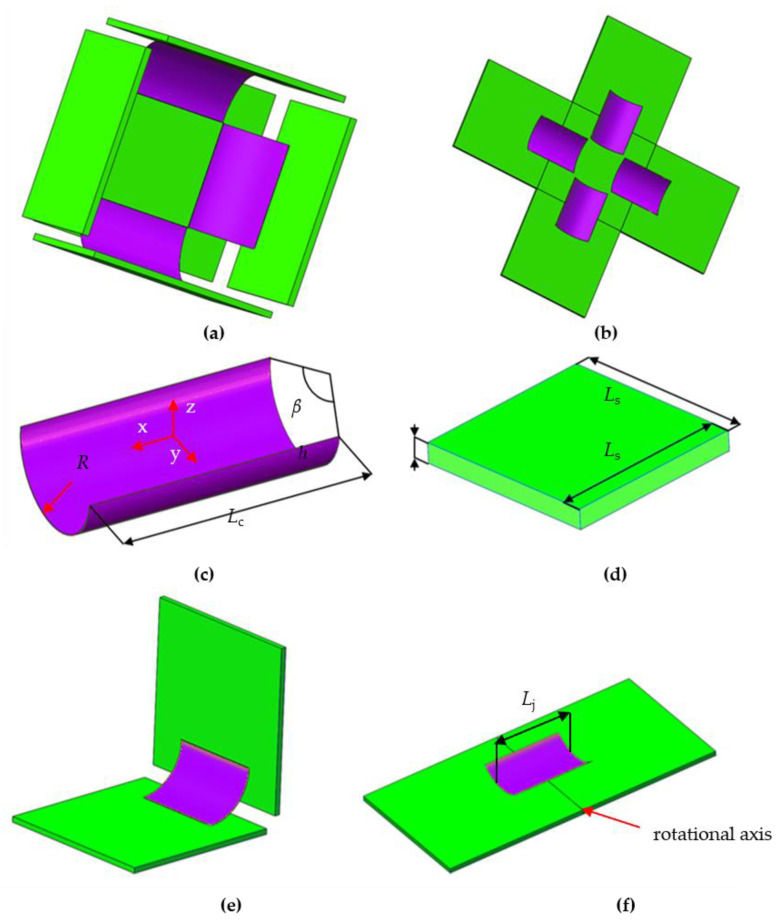
Three-dimensional model of the deployable panel model. (**a**) whole model; (**b**) core components of the deployable panel model; (**c**) cylindrical shell; (**d**) deployable panel; (**e**) folded configuration of one quarter of the deployable panel model; (**f**) flattened configuration of one quarter of the deployable panel model.

**Figure 2 materials-15-01942-f002:**
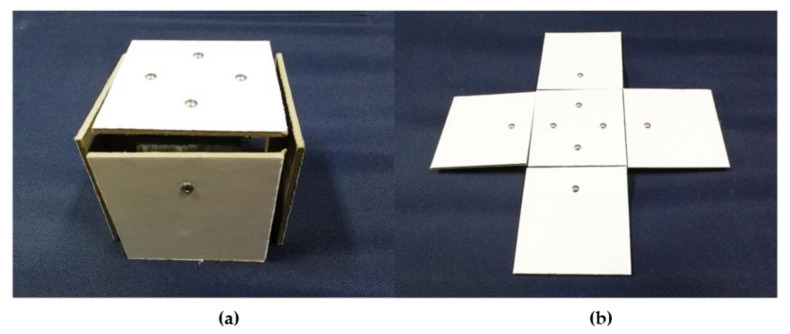
Different stable configurations of the specimen. (**a**) folded configuration; (**b**) flattened configuration.

**Figure 3 materials-15-01942-f003:**
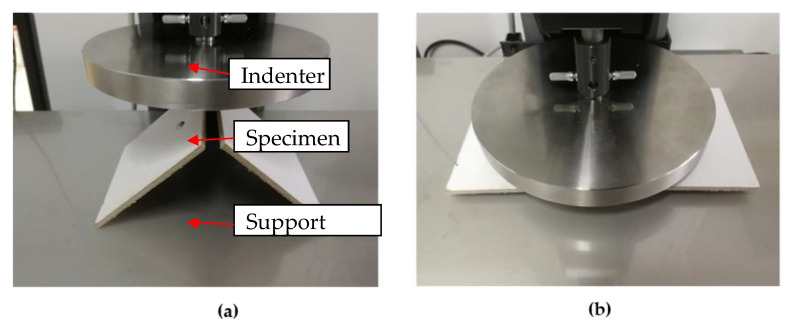
Experimental test platform. (**a**) Folded configuration; (**b**) flattened configuration.

**Figure 4 materials-15-01942-f004:**
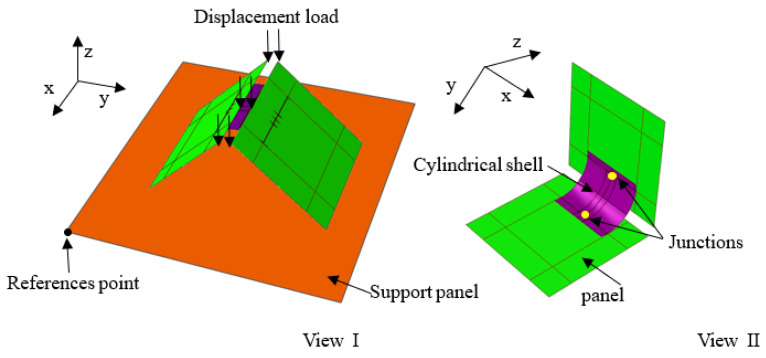
Finite element model of one quarter of the deployable panel model with bistable cylindrical shells.

**Figure 5 materials-15-01942-f005:**
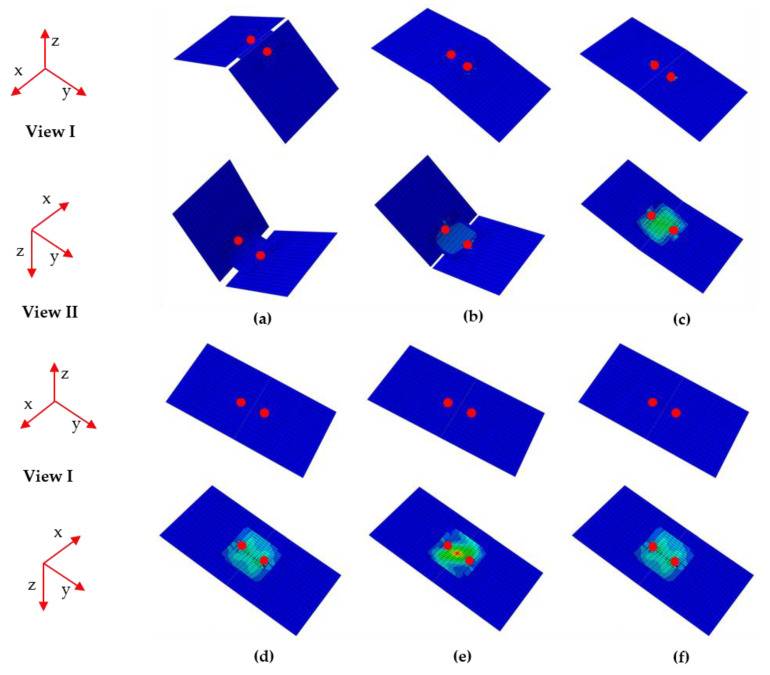
Deformation process of one quarter of the deployable panel model. (**a**) Folded configuration; (**b**) starting to deform; (**c**) starting to snap through; (**d**) flattened configuration; (**e**) starting to recover; (**f**) flattened configuration.

**Figure 6 materials-15-01942-f006:**
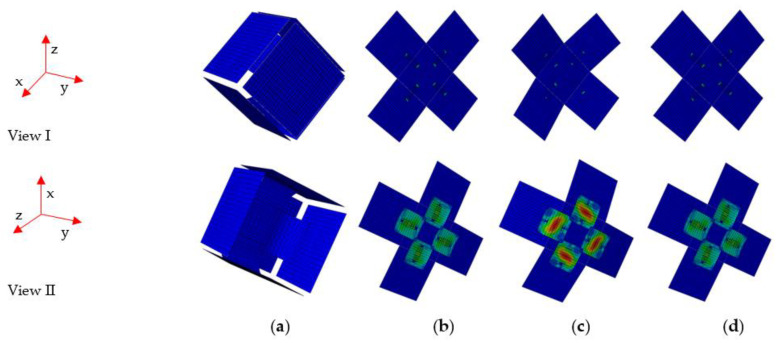
Deformation process of the entire deployable panel model. (**a**) Folded configuration; (**b**) unfolding state; (**c**) unstable state; (**d**) flattened configuration.

**Figure 7 materials-15-01942-f007:**
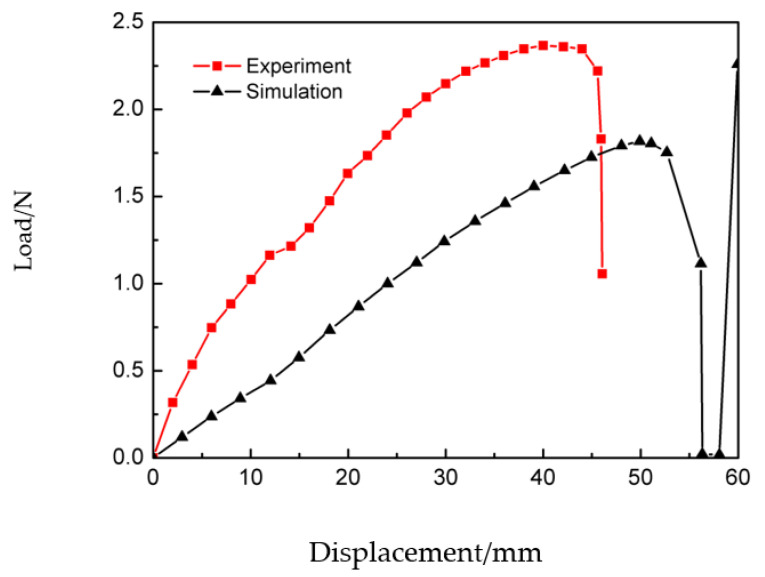
Simulated and experimental load–displacement curves of the deployable panel model.

**Figure 8 materials-15-01942-f008:**
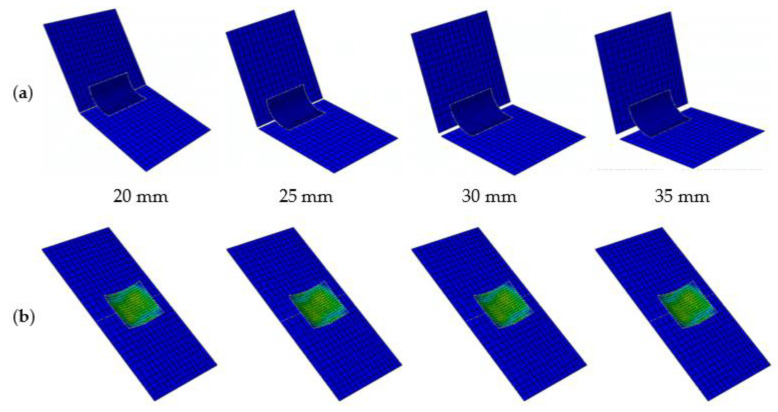
Different stable configurations of the deployable panel model in simulations with different values of *L*_j_. (**a**) folded configuration; (**b**) flattened configuration.

**Figure 9 materials-15-01942-f009:**
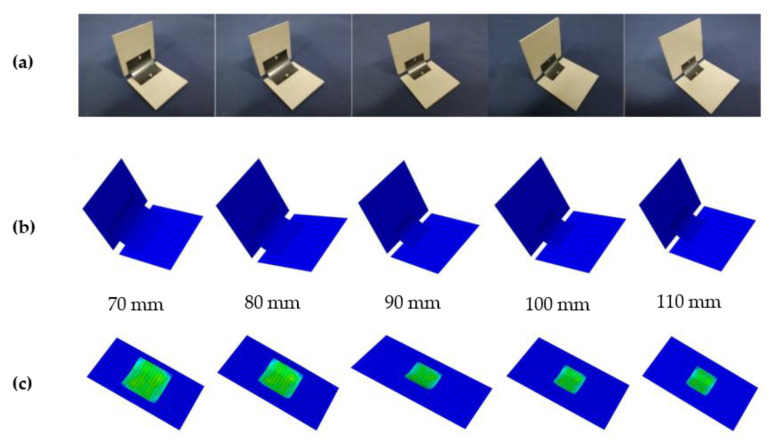
Different stable configurations of the deployable panel model in simulations with different dimensions. (**a**) Specimens of folded configuration; (**b**) finite element models of folded configuration; (**c**) finite element models of flattened configuration.

**Figure 10 materials-15-01942-f010:**
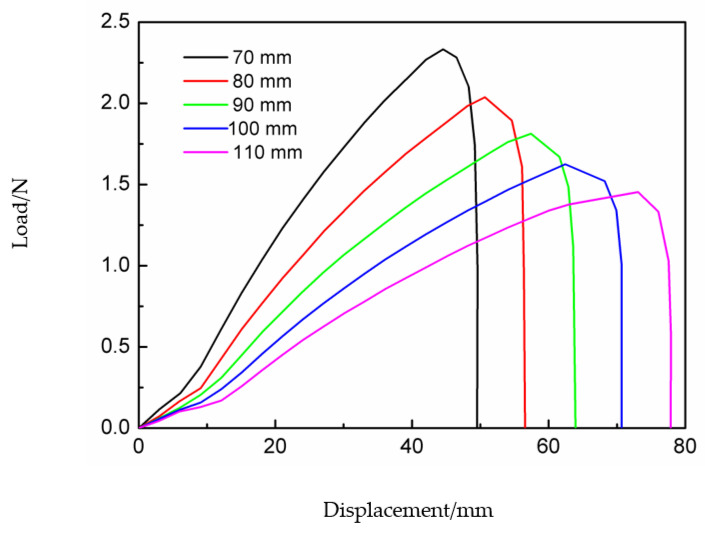
Load–displacement curves of the deployable panel model with different sizes in the simulation.

**Figure 11 materials-15-01942-f011:**
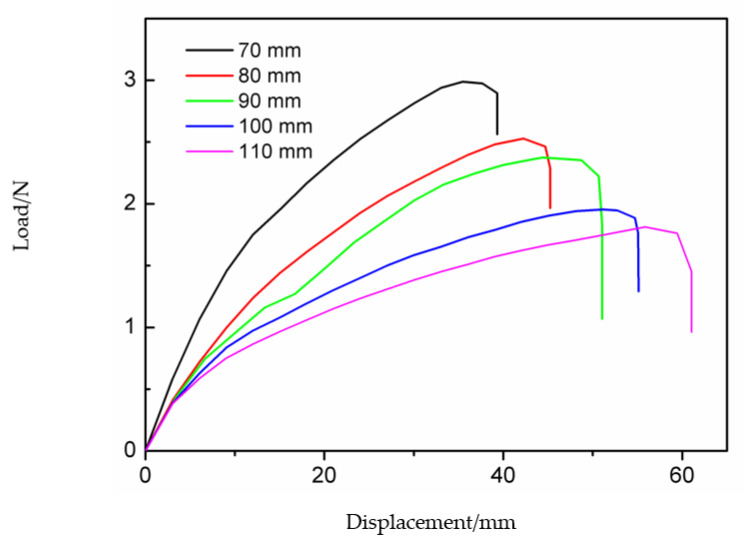
Load–displacement curves of the deployable panel model specimen with different sizes in the experiment.

**Figure 12 materials-15-01942-f012:**
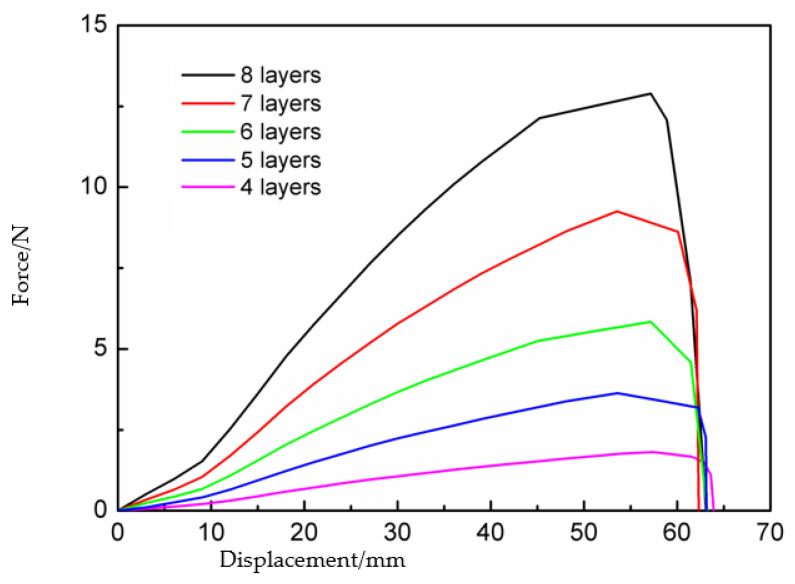
Load–displacement curves of the deployable panel model in simulations with different numbers of layers of the cylindrical shell.

**Table 1 materials-15-01942-t001:** Material properties of the cylindrical shell.

Longitudinal Modulus *E*_11_ (GPa)	Transverse Modulus *E*_22_ (GPa)	Shear Modulus in the 1–2 Plane *G*_12_ (GPa)	Shear Modulus in the 1–3 Plane *G*_13_ (GPa)	Shear Modulus in the 2–3 Plane *G*_23_ (GPa)	Poisson Ratio *ν*_12_	Single Layer Thickness *t* (mm)
132	10.3	6.5	6.5	3.9	0.25	0.12

**Table 2 materials-15-01942-t002:** Geometric parameters of the deployable panel model.

Length of the Cylindrical Shell *L_c_* (mm)	Central Angle *β* (°)	Radius *R*(mm)	Sidelength of the Deployable Panel *L_s_* (mm)	Thickness of the Deployable Panel *h* (mm)	Distance between Two Junctions *L*_j_ (mm)
50	90	25	90	3	39.25

**Table 3 materials-15-01942-t003:** Main technical parameters of the deployable panel model with different distances between two Junctions.

Distance between Two Junctions *L*_j_ (m)	Coverage Area *a* (m^2^)	Package Volume *v* (m^3^)	Ratio of Coverage Area and Package Volume *μ* (m^−1^)
20 × 10^−3^	405 × 10^−4^	1252 × 10^−6^	32.35
25 × 10^−3^	405 × 10^−4^	1270 × 10^−6^	31.89
30 × 10^−3^	405 × 10^−4^	1171 × 10^−6^	34.59
35 × 10^−3^	405 × 10^−4^	963 × 10^−6^	42.06
39.5 × 10^−3^	405 × 10^−4^	729 × 10^−6^	55.56

**Table 4 materials-15-01942-t004:** Main technical parameters of the deployable panel model with different panel sidelengths.

Sidelength of the Deployable Panel *L*_s_ (m)	Coverage Area *a* (m^2^)	Package Volume *v* (m^3^)	Ratio of Coverage Area and Package Volume *μ* (m^−1^)
70 × 10^−3^	245 × 10^−4^	343 × 10^−6^	71.43
80 × 10^−3^	320 × 10^−4^	512 × 10^−6^	62.50
90 × 10^−3^	405 × 10^−4^	729 × 10^−6^	55.56
100 × 10^−3^	500 × 10^−4^	1000 × 10^−6^	50.00
110 × 10^−3^	605 × 10^−4^	1331 × 10^−6^	45.45

## Data Availability

Not applicable.

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
