# Peer review of "Novel Deployable Panel Structure Integrated with Thick Origami and Morphing Bistable Composite Structures"

_materials, 2022, doi:10.3390/ma15051942_

Round 1

Reviewer 1 Report

  • Why did you use the quarter panel for the experimental test? Since you simulated both the complete and the quarter panel, you could present a comparison between both to see how the experimental data will compare to the actual application you propose.
  • Please expand the discussion: why does the structure have a bistable behavior? what parameter has the greatest influence on the critical load? You could provide design guidelines to achieve an optimal design. The goal is not clear; should the critical load be maximized or minimized?
  • Please state the future work that could come from this manuscript. You could also provide a better justification for using this structure in solar panels, as well as further applications.
  • The numbering of the Conclusion section is incorrect, please update.
  • Check the layout of the figure elements, especially in Figures 1, 2, 3, 9, and 11. Consider saving the whole figures as separate image files for their insertion in the manuscript.
  • Figure 7 is missing a label in the vertical axis.
  • Too short section of conclussions, expand and discuss in more detail. 

This couple of sentences seem to suggest that the work is still in progress and not ready for publication. First a good agreement is not enough for validation, maybe not the proper use of words. Second, it is suggested to solve the problems and redo the tests. 

"The results of the numerical simulation and experimental test were compared and were found to be in good agreement. Moreover, the reasons that may cause the errors and the factors that affect the trigger force and coverage area-to-package volume ratio of the deployable panel model were discussed."

Reviewer 2 Report

In this manuscript, a deployable panel structure with flexible materials and thick panels is proposed and the experimental and numerical verifications are conducted.

The authors explained that the proposed structure is based on the ORIGAMI technique; however, it is considered to be different from origami, since the proposed structure is assembled by just sticking different type of materials.

It should be clarified what features of origami are utilized in this method.

The verified results could be considered a vary natural results. It should be considered how these results can be useful in design and application.

In Fig. 1, some characters displaying the parameters are misaligned or missing.

In the experimental verifications, details of the equipment used in the experiments should be provided.

Reviewer 3 Report

The paper is interesting to read and it is easy to understand what is your goal. I think this large number of references is not well-founded. There are many similar works.

The presentation of your results need some improvement to eliminate the missing or misplaced labels. What are shown in the screenshots coming from Abaqus? What do the colors represents?

Please do some spellchecking.

Reviewer 4 Report

The manuscript entitled "Novel Deployable Panel Structure Integrated with Thick Origami and Morphing Bistable Composite Structures" presents a deployable panel structure inspired by the thick origami structure and bistable composite cylindrical shell that can remain in either a folded or a flattened state without the need for continuous external loading. The goals and motivation of study are clear. The work is interesting and pleasant, and clearly shown. The results are a promising and exciting area for solar panels applications. The paper can be recommended to Materials.

Additional comments that need to be addressed:

  1. The sentence: In addition, finite element simulations were carried out and then compared with experimental results.' should be In addition, finite element simulations were carried out and compared with experimental results.'
  2. The sentence: 'The factors affecting the trigger force and load–displacement curve of the deployable structure were discussed.' should be 'The factors affecting the deployable structure's trigger force and load-displacement curve were discussed.'
  3. The sentence: 'Both experimental and numerical results show that the proposed deployable panel model can remain either a folded (spatial) state or an unfolded (flatten) state, which sheds light on the design of spatial deployable structures.' should be Both experimental and numerical results show that the proposed deployable panel model can remain either a folded (spatial) state or an unfolded (flatten) form, which sheds light on the design of space deployable structures.'
  4. Figures 10 and 11 should be corrected because they cover up the caption of figures. Moreover, the caption of figure 11 is written in a different font.

Reviewer 5 Report

This paper proposes an interesting approach to deployment. The bi-stability of cylindrical shells is exploited for deployment.

There are a few serious issues that need to be addressed before any further consideration.

  1. The contact points between the facets and the shells are not clear. The contact points should be clearly shown at the two stable stages and one intermediate stage.
  2. The curves in fig 7 do not really agree
  3. While the authors claim that the bistable shells are made of CFRP composites, from some of the figures of physical experiments these appear to be made of metal. please clarify.
  4. More works on origami and folding should be cited (such as: Applied Materials Today, 20 100715) for giving a broader perspective of the field.
  5. One of the major lacunae of this work seems to be the fact that the the two connected facets will not make a 90 degree angle at one of the stable stages. To complete the cube for many applications of deployable structures, it is necessary. The authors need to clarify/ address this issue.

Round 2

Reviewer 2 Report

The manuscript is considered to be almost revised in response to review comments. 

Reviewer 5 Report

accept